# Are Trauma Surgery Simulation Courses Beneficial in Low- and Middle-Income Countries—A Systematic Review and Meta-Analysis

**Yousuf Hashmi** [1,2,*] [iD]**, Nashmeeya Ayyaz** [2]**, Hamza Umar** [2]**, Anam Jawaid** [2] **and Zubair Ahmed** [1,3,4,*] [iD]

1    Institute of Inflammation and Ageing, University of Birmingham, Edgbaston, Birmingham B15 2TT, UK
2    College of Medical and Dental Sciences, University of Birmingham, Edgbaston, Birmingham B15 2TT, UK; nxa669@student.bham.ac.uk (N.A.); hxu610@student.bham.ac.uk (H.U.); axj734@student.bham.ac.uk (A.J.)
3    Surgical Reconstruction and Microbiology Research Centre, National Institute for Health Research, Queen Elizabeth Hospital, Birmingham B15 2TH, UK
4    Centre for Trauma Sciences Research, University of Birmingham, Edgbaston, Birmingham B15 2TT, UK
\*    Correspondence: ysh648@student.bham.ac.uk (Y.H.); z.ahmed.1@bham.ac.uk (Z.A.)

**Abstract:** Despite trauma-related injuries being a leading cause of death worldwide, low- and middle-income countries (LMICs) lack the infrastructure and resources required to offer immediate surgical care, further perpetuating the risk of morbidity and mortality. In high-income countries, trauma surgery simulation courses are routinely delivered to surgeons, teaching the fundamental skills of operative trauma. This study aimed to assess whether similar courses are beneficial in LMICs and how they can be improved. We performed a systematic review and meta-analysis using MEDLINE, Embase and Google Scholar, analysing studies evaluating trauma surgery simulation in LMICs. The outcomes measured included clinical knowledge improvement, participant confidence and general course-feedback. The review was carried out in-line with PRISMA guidelines. Five studies were included, summating a population of 172 participants. In three studies, meta-analysis showed an overall significant weighted mean improvement of knowledge post-course by 22.91% (95%CI 19.53, 26.29; $p < 0.00001$; $I^2 = 0\%$). One study reported a significant increase in participant confidence for 20/22 of operative skills taught ($p < 0.04$). We conclude that these courses are beneficial in LMICs; however, further research is necessary to establish the optimum course design, and whether patient outcomes are improved following their implementation. Collaboration between international trauma institutions is essential for closing the educational resource inequality gap between higher- and lower-income countries.

**Keywords:** trauma; surgery; simulation; medical education

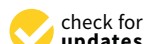



## 1. Introduction

Trauma-related injuries requiring surgical care cause approximately 16.9 million deaths annually, more than HIV, malaria and tuberculosis combined [1,2]. Low- and middle-income countries (LMICs) are the worst affected, accounting for 90% of these deaths [3]. Whilst a major reason for this is due to poor primary prevention such as road safety protocols, inadequate pre-hospital and hospital care are also accountable. High-income countries have implemented major trauma centres (MTCs) and networks that have drastically reduced patient mortality from trauma-related injuries [4]. These networks have excellent transport links and highly efficient triage systems in place, ensuring every trauma patient is able to receive optimum treatment by 'hyper-specialized' surgical teams. These systems are simply not available in most LMICs due primarily to resource scarcity, a lack of infrastructure and a workforce in short supply [5]. It is estimated that approximately 2 million lives can be saved annually if LMICs had a similar level of trauma care compared with higher income countries [6].

Simulation training involves the artificial replication of clinical scenarios in order to reduce errors when completing procedural tasks in the real world. Studies have shown that simulating basic surgical skills, such as suturing, results in improved surgeon performance when in the operating room [7]. The use of simulation has also become fundamental to trauma education in high-income countries, with trauma clinicians routinely attending specific advanced trauma training courses such as the Advanced Trauma Life Support (ATLS) course [8]. Multiple assessments of trauma units in LMICs have identified inadequacies in staff knowledge and execution of clinical scenarios, recommending an increase in staff education via courses such as ATLS [9–11]. Whilst this course represents the gold-standard in high-income countries, its implementation in LMICs is restricted due to several of the recommendations of the course utilizing resources that are not readily available in LMIC trauma units [12]. This has led to the subsequent development of primary trauma care courses (PTCs), which have adapted the fundamental principles of the ATLS into a sustainable programme tailored to the resource-limited environment of LMICs. A systematic review by Kadhum et al. [13] assessed the benefit of PTCs in LMICs and found that despite further high-quality research being needed, these courses may improve the clinical knowledge and competence of participants.

However, PTCs predominantly focus on basic resuscitation and emergency care simulation for medical doctors and pre-hospital staff. There remains a deficiency in LMICs of high-quality operative courses for trauma surgeons, which is alarming considering the importance of definitive surgical treatment in reducing the morbidity and mortality of trauma patients. Whilst high-income countries are able to run surgical trauma courses such as the American College of Surgeons Advanced Surgical Skills for Exposure in Trauma costing up to $2000 per student [14], LMICs do not have the same luxury. Thus, there is a clear need for a sustainability assessment of trauma surgery courses in LMICs, to establish recommendations for their improvement and widespread implementation.

The objective of this systematic review was to assess the effectiveness of trauma surgery simulation courses in improving the clinical and operative knowledge of surgeons in LMICs. Secondary objectives include whether these courses cause an improvement in the following: self-rated confidence and competence of surgeons; participant experience via satisfaction scores and qualitative survey response; patient outcomes and whether any improvements are sustained over time. Following an evaluation of the literature, recommendations for the improvement of trauma simulation courses in LMICs are discussed.

## 2. Materials and Methods

### 2.1. Literature Search

This systematic review was conducted in accordance with the Preferred Reporting Items for Systematic Reviews and Meta-Analysis (PRISMA) group guidelines [15]. Two reviewers (Y.H., N.A.) systematically searched PubMed, MEDLINE and Embase electronic databases utilizing the OVID search interface from inception to 5th July 2021. The Google Scholar database was also reviewed. We used a range of search terms and applied to all fields, rather than just the title, to ensure a thorough search. Boolean operators were utilised as follows: ("Trauma" [TEXT] AND "Surg*" AND "Skill*" [TEXT] AND ("Simulation" [TEXT] OR "Programme" [TEXT] OR "Course" [TEXT])). Full search strategy results are provided in File S1.

### 2.2. Eligibility Criteria

The inclusion and exclusion criteria for our review is provided in Table 1. We did not restrict our search based on sample size or language. No date restrictions for studies were implemented, with all studies published before the date of search (5 July 2021) eligible for inclusion. Any disagreements on study inclusion were resolved by discussion with the senior author (Z.A).

**Table 1.** Eligibility Criteria.

| PICOS | Inclusion Criteria | Exclusion Criteria |
|---|---|---|
| Population | Surgeons<br>Surgical trainees | Medical students, non-surgical doctors, and other allied health specialties such as nurses, midwives, and physiotherapists. |
| Intervention | Trauma surgery simulation courses conducted in low- or middle-income countries as defined by The World Bank Classification [16] | Non-trauma related surgical courses (e.g., elective orthopaedic surgery courses, general surgery courses etc.)<br>Courses with no operative tasks. |
| Comparison | N/A | N/A |
| Outcome | Primary outcome: clinical/operative knowledge improvement<br>Secondary outcomes: general course feedback scores, immediate and long-term self-rated confidence scores | N/A |
| Study design | Primary data studies including randomised controlled trials and observational studies | Reviews, abstracts, case reports or quality improvement projects |

### 2.3. Study Selection

Two authors (Y.H. and N.A.) independently screened the total list of studies retrieved by the literature search initially by title and abstract. Following this initial screen, the remaining studies underwent a full-text analysis. In addition, Y.H. examined the full reference lists of selected studies to identify additional studies.

### 2.4. Data Extraction

Y.H. and A.J. completed a data extraction spreadsheet for this review, with any disagreements reviewed by the senior author (Z.A). The following data were extracted from all included studies: location, study type, participant number as well as their career stage and demographic data. Data were also extracted on the courses evaluated in each study including course design, duration, content and cost of implementation. We contacted corresponding authors via email to request any additional information if available.

The primary outcome extracted for this review was improvement in clinical and operative knowledge of participants following the trauma surgery course. This is most commonly via pre- and post-course test scores, reported as mean $\pm$ standard deviation. This was selected as the primary outcome as it is an objective measure, unlike self-reported confidence scores, and is commonly recorded in medical education primary studies. Secondary outcomes extracted include general course feedback scores, immediate and long-term self-rated confidence scores as well as any other relevant survey data.

### 2.5. Risk of Bias

We assessed the risk of bias in the included studies using the risk of bias in non-randomized studies of interventions (ROBINS-I) tool [17]. Two reviewers (Y.H. and H.U.) independently applied a risk of bias judgement score (low, moderate or high) for each of the tool's seven specific domains. The overall risk of bias for a study was determined by the highest risk judgement score in any of the seven assessed domains. Any disagreements were resolved through discussion with the senior author (Z.A).

### 2.6. Statistical Analysis

Data are presented as mean $\pm$ standard deviation (SD). Individual and combined average pre- and post-test scores were analysed in SPSS, Version 20 (IBM Corporation, Chicago, IL, USA) using a two-tailed paired sample *t*-test. *p* values < 0.05 were considered significant. Despite the differences in methods used to evaluate knowledge, the heterogeneity of participants and the small sample size, we performed a meta-analysis using

Review Manager 5.4.1 (Cochrane Informatics and Technology), employing the random effects model.

## 3. Results

### 3.1. Study Selection

The initial search from all databases yielded 880 results, with two further studies identified from other sources. Following the removal of duplicates, 608 studies were extracted into a database for screening, 531 were excluded due to the pre-stated inclusion and exclusion criteria, resulting in 77 for full-text analysis. A further 72 studies were then excluded after a full-text search, resulting in the five primary studies included in this systematic review. A list of full-text exclusions (n = 72) is provided in File S2. The full PRISMA flow chart, including reasons for exclusion, is provided in Figure 1.

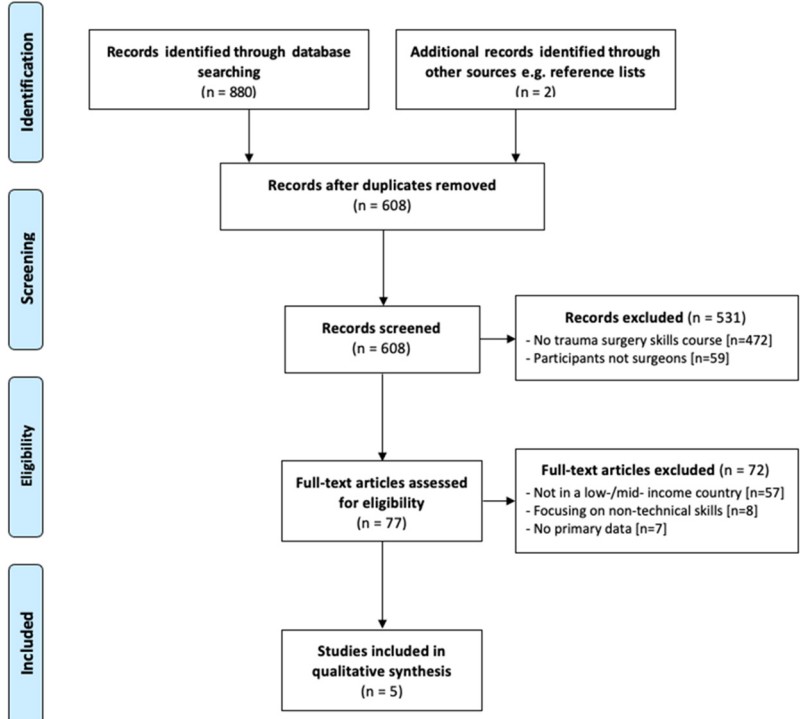

**Figure 1.** PRISMA flow diagram.

### 3.2. Study Characteristics

All five included studies were published between June 2005 and December 2020 and were undertaken in four different, low- or middle-income countries: Uganda, Pakistan, Brazil and Ghana. Three studies [18–20] were non-randomized cross-sectional studies and one was a randomized prospective crossover study [21]. Jacobs et al. [22] had an unspecified study design and no raw outcome data. All studies evaluated different courses: the Mulago Operative Trauma Resuscitation Course (M-OTR) [18], Surgical Techniques and Repairs in Trauma for the Low-resource Environment (STaRTLE) [19], Emergency Vascular Surgery Course for Non-Vascular Surgeons (EVSC) [20], Advanced Trauma Life Support (ATLS) surgical skills station [21] and the Advanced Trauma Operative Management (ATOM) course [22]. A detailed description of each course is provided in Table 2. A total of 172 participants attended the courses, with 114 surgical-trainees and 58 at a senior level.

**Table 2.** Course Descriptions.

| Study | Course | Course Format | Course Length | Course Content | Cost Description |
|---|---|---|---|---|---|
| Ullrich et al., 2020 [18] | M-OTR | Didactic lectures. Practical sessions, daily cadaver lab sessions. | 3 days | Team management dynamics. Primary and secondary survey. Airway management. Ultrasound uses in trauma. Trauma radiology. Penetrating + blunt injuries to the neck, chest, abdomen, and pelvis. | $5000 initial investment and $1500 per course. $186 charged per student. |
| Anderson et al., 2018 [19] | STaRTLE | Didactic lectures. Practical sessions, cadaver-based surgical technique education. | 2 days | Operative techniques in the chest, abdomen, neck and extremities. | Unspecified (low). |
| Rehman et al., 2020 [20] | EVSC | Interactive lectures. Video demonstrations. Practical vascular skills training on animal models. | 1 day | Common vascular emergencies. Vessel exposure. Arteriotomy and primary closure. End-to end anastomosis. Shunt placement. Performing embolectomy. Performing fasciotomy. | Unspecified. |
| Garcia et al., 2019 [21] | ATLS (surgical skills stations) | 60 min per station (n = 3). Procedures simulated on TraumaMan, SurgeMan and live animals. | 1 day | Cricothyroidotomy Tube thoracostomy Pericardiocentesis Diagnostic peritoneal lavage | TraumaMan simulator; $30,000 initial investment + $6000 per course. SurgMan simulator; $2500 initial investment + $650 per course. |
| Jacobs et al., 2005 [22] | ATOM | Didactic lectures (n = 6). Practical operative skills sessions on 50 kg swine with 12 pre-created standardised injuries. | 1 day | Injuries to various organ systems including trauma to the bowel, bladder, ureter, kidney, duodenum, pancreas, liver, stomach, spleen, diaphragm, inferior vena cava, and heart. | Unspecified (high). |

Notes: M-OTR: Mulago Operative Trauma Resuscitation Course; STaRTLE: Surgical Techniques and Repairs in Trauma for the Low-resource Environment; EVSC: Emergency Vascular Surgery Course for Non-Vascular Surgeons; ATLS: Advanced Trauma Life Support (ATLS); ATOM: Advanced Trauma Operative Management Course.

Three of the studies [18–20] reported our primary outcome measure of knowledge improvement via pre- and post-course tests. Table 3 provides an overview of the study characteristics including all outcome measures assessed.

**Table 3.** Study Characteristics.

| Study | Course | Location | Study Design | Participants | Outcome Measures |
|---|---|---|---|---|---|
| Ullrich et al., 2020 [18] | M-OTR | Kampala, Uganda | Non-randomised cross-sectional study | 52 surgical trainees | Knowledge improvement via pre- and post-course test (n = 48). General course review survey. General trauma education needs assessment survey (n = 28). Resource utilisation survey (n = 18). |
| Anderson et al., 2018 [19] | STaRTLE | Mbarara, Uganda | Non-randomised cross-sectional study | 8 surgical trainees | Knowledge improvement via pre- and post-course test (n = 8). Participant operative skill confidence via pre- and post-course survey (n = 8). Long-term operative skill confidence via 1–2 month (n = 8) and 1-year survey (n = 4). |
| Rehman et al., 2020 [20] | EVSC | Karachi, Pakistan | Non-randomised cross-sectional study | 21 total participants, 18 surgical trainees and 3 consultant surgeons | Knowledge improvement via pre- and post-course test (n = 21). General course review survey (n = 21). |
| Garcia et al., 2019 [21] | ATLS (surgical skills stations) | Sao Paulo, Brazil | Randomised prospective crossover study | 36 surgical trainees | User satisfaction of SurgeMan, TraumaMan and animal models for use in surgical skills station via survey (n = 36). |
| Jacobs et al., 2005 [21] | ATOM | Accra, Ghana | n/a | 55 surgeons | n/a |

Notes: M-OTR: Mulago Operative Trauma Resuscitation Course; STaRTLE: Surgical Techniques and Repairs in Trauma for the Low-resource Environment; EVSC: Emergency Vascular Surgery Course for Non-Vascular Surgeons; ATLS: Advanced Trauma Life Support (ATLS); ATOM: Advanced Trauma Operative Management Course.

*3.3. Results of Individual Studies—Primary Outcome*

All three studies assessing our primary outcome of knowledge improvement reported a statistically significant increase in post-course test scores. For example, the average test score (mean ± standard deviation) improved from 55.4 ± 13.9% to 78.1 ± 11.6% ($p < 0.001$ (paired *t*-test)). The individual mean test scores and the combined scores of all three studies are presented in Table 4.

**Table 4.** Test Scores (Mean ± Standard Deviation).

| Study | Course Assessment | Participants | Mean Pre-Test Score (% ± SD) | Mean Post-Test Score (% ± SD) | *p*-Value |
|---|---|---|---|---|---|
| Ullrich et al., 2020 [18] | Written exam | n = 48/52 | 56.0 ± 10.0 | 79.0 ± 9.0 | <0.05 |
| Anderson et al., 2018 [19] | 20-item MCQ | n = 8/8 | 50.7 ± 10.5 | 73.6 ± 9.1 | 0.002 |
| Rehman et al., 2020 [20] | 20-item MCQ | n = 21/21 | 59.5 ± 21.3 | 81.6 ± 16.6 | <0.001 |
| Combined | Written exam/MCQ | n = 77/81 | 55.4 ± 13.9 | 78.1 ± 11.6 | <0.001 |

Despite the limited number of studies, we conducted a meta-analysis which showed that there was an overall significant weighted mean improvement of knowledge by 22.91% (95%CI 19.53, 26.29; $p < 0.00001$; $I^2 = 0\%$) after taking part in the relevant trauma surgery course (Figure 2).

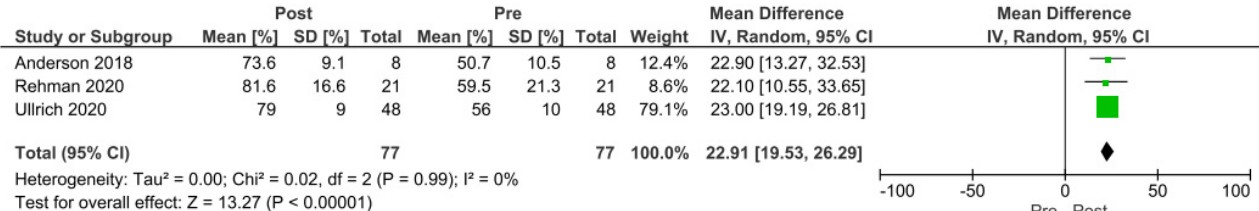

**Figure 2.** Forest plot of knowledge improvement after (Post) participation in a relevant trauma surgery simulation course. Green squares = weighted mean difference. Black diamond = overall weighted mean difference.

### 3.4. Results of Individual Studies—Secondary Outcomes

Ullrich et al. [18] evaluated the M-OTR course across three years from 2017 to 2019 in Uganda. From the course feedback survey, all participants believed that the course increased their knowledge, with 96% feeling that they would be able to teach the course content to others. A needs assessment survey was distributed during the first two years of the course to determine participants' view on trauma education as a whole. A total of 79% of surgical trainees agreed that their trauma operative skills needed the most improvement compared with other surgical skills; 71% were unaware of surgical training opportunities in Uganda outside their curriculum.

The study by Anderson et al. [19] evaluated the STaRTLE course in Uganda. Immediate pre-course and post-course surveys demonstrated a statistically significant ($p < 0.04$) improvement in trainee comfort level for 20 of the 22 operative skills taught. Follow-up surveys highlight that trainee comfort in operative skills was sustained at 1–2 months and 1 year after the course. Rehman et al. [20] described the effectiveness of the EVSC vascular skills workshop and 90.5% of the participants rated the workshop as 'excellent' or 'very good'.

Garcia et al. [21] evaluated the use of TraumaMan, SurgeMan and animal models for use in the ATLS surgical skills simulation. TraumaMan is a high-cost artificial manikin commonly used in high-income countries and was rated significantly better than the more affordable SurgeMan ($p < 0.05$) and animal models ($p < 0.05$). When participants were questioned which of the three models they would recommend for the course if no financial or ethical considerations were present, 62% chose the animal model, 30% chose TraumaMan and 8% chose the SurgeMan model.

Whilst Jacobs et al. [22] had no raw quantitative data, it was reported that within a week of completing the ATOM course, one of the participants managed a patient with a severe gunshot wound to the liver utilising a technique taught directly during the course.

### 3.5. Risk of Bias in Studies

The four non-randomized studies were assessed across all seven domains for the potential risk of bias in the methodology and outcomes (Figure 3). The overall bias of all included studies was found to be a moderate risk of bias due to the potential of confounding variables influencing the outcomes. The studies failed to include an appropriate regression, standardisation, or matching method to account for this risk. No risk of bias analysis was performed for Jacobs et al. [22] due to the absence of a study design or raw outcome data.

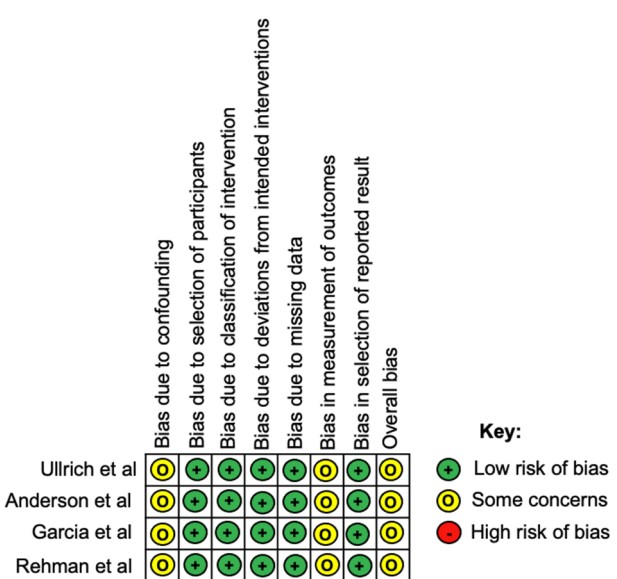

**Figure 3.** Risk of bias assessment according to the ROBINS-I tool.

## 4. Discussion

Globally, trauma education is poorly delivered in comparison with other more traditional surgical specialties such as orthopaedics, neurosurgery and urology [23,24]. This is particularly true in LMICs, where a deficiency of trauma-related education and simulation has resulted in the vast majority of frontline healthcare workers caring for injured patients with no formal training in trauma [25,26]. As discussed previously, the emphasis of trauma educational initiatives thus far has been on PTCs rather than trauma surgery simulation courses. With an estimated 21% of the trauma burden in LMICs deemed "avertable" following the provision of basic surgical trauma care, it is essential that more operative courses are facilitated in these areas [27]. Evaluation of surgical skills courses are well-established in the literature; however, this is usually via general surgical skills courses rather than those specific to trauma surgery [28,29]. Unsurprisingly, 79% of participants in the study by Ullrich et al. [18] agreed that their trauma operative skills needed the most improvement.

The aim of this systematic review was to ascertain whether completion of trauma surgery simulation courses is beneficial to trauma care in LMICs. The results of our primary outcome analysis suggest that these courses improve participants' clinical and operative knowledge, which is encouraging. These courses also improve surgeons' confidence in performing operative skills, allowing them to have a more systematic approach when dealing with complex cases [19]. Participants rate these courses highly in post-course surveys, and they retain the principles taught in the course for their own clinical practice. However, the lack of recorded patient outcomes mean that we are unable to establish whether these courses result in improved patient care or whether they have any effect on patient morbidity or mortality.

The difficulty in evaluation of trauma courses is not unique to LMICs. A review by Mackenzie et al. [14] identified 21 trauma skills courses, the majority delivered in high-income countries, and concluded that these programmes are variable in their design, content, duration, cost and resource requirements. In addition, the lack of universal performance metrics and outcomes makes determining the success of these courses difficult. These findings are supported by our review, where each of the five included studies evaluated their course differently. Until these objective performance metrics are established, the most efficient and effective method of trauma skills course design will remain unknown, as they are essential when comparing course efficacies in different studies [30–32]. Despite the heterogeneity, a meta-analysis of the test scores before and after the relevant course demonstrated an overall significant improvement in knowledge, indicating the benefits of

such courses. However, there are only three studies with a total of 77 participants, which is too small to make definitive conclusions about the benefits of such courses.

A standardised training course offers a potential solution, ensuring all surgeons are operating to an internationally pre-defined standard. However, delivering surgical courses in LMICs as they are currently designed in high-income countries is challenging. It was clear from our results that adaptations have to be made to courses in order to successfully run in a low-resource setting; Ullrich et al. [18] improvised the use of gowns for abdominal packing due to the lack of laparotomy pads. Moreover, course materials for trauma education courses in high-income countries are not freely offered online, which is disappointing considering the wealth and resource inequality between countries. Course designers in LMICs are forced to "reinvent the wheel" when establishing programme content and scenarios [18]. Fortunately, the M-TOR course described in the study by Ullrich et al. [18] is due to be open source with free distribution of course materials; this may encourage more course developers to share resources. Ultimately, greater collaboration between high- and lower-income countries is essential in improving the quality of trauma care globally.

A successful example of this collaboration is described in the article by Jacobs et al. [22] which showcases the implementation of the ATOM course in Ghana. This course was designed in collaboration with the American College of surgeons to improve the training of trauma surgeons in the area. Financial aid was donated by Johnson & Johnson in order to provide resource and equipment support. The article highlights that despite the resources and initial delivery of the course being pioneered by the high-income organisation, ultimately a sense of ownership must be established by the host institution [22]. This develops a sense of confidence and pride by the local team, thus ensuring a sustainable system that achieves the same standards as teaching centres in high-income countries [22].

Another important consideration is the background of participants undertaking the course. Surgeons in high-income countries such as the United States have been educated via traditional didactic lectures followed by operative skills practice. Unsurprisingly, the vast majority of courses are therefore structured in this way as they have been adapted from the ATLS or ATOM programmes. However, surgeons from LMICs may be more familiar with different teaching methods throughout their medical school and higher specialty training [33]. There must be transparent communication between course organisers and host training facilities when designing the curriculum for these courses. This recommendation is supported by Anderson et al. [19], where a close relationship with the course developers and solid understanding of the resources available were pivotal in the success of the programme.

The impact of non-technical factors such as leadership skills and communication on patient outcomes is another challenge when evaluating a surgeon's clinical improvement following a skills course [34]. Our review only evaluated courses with an element of operative skills teaching targeted at surgeons. There is a deficiency in the literature of courses teaching multi-disciplinary skills to all members of the surgical team including anaesthetists, surgical nurses, and surgeons collaboratively. As the final outcome of the patient is unlikely due solely to the operative skill of the lead surgeon, further studies are needed delivering high-quality education on communication and other non-technical skills between team members in high-pressured simulated environments. Moreover, trauma patients are managed by a full multi-disciplinary team including pre-hospital care, surgical care as well as rehabilitation post-operatively. Rather than focussing solely on operative management from surgeons, it is critical to improve all stages of trauma patient care to improve patient outcomes.

Importantly, simply attending an educational trauma course does not always correspond to better clinical practice. Participants in a study by Cioè-Peña et al. [35] did not effectively acquire the principles taught on the PTC. A follow-up study identified one reason for this being 'mentally absent' participants who had attended following a night shift. Some candidates were not even present for the entire course due to clinical duties;

this issue is less common in high-income countries where designated time-off to attend training courses is often received by staff [36]. Multiple factors must be considered when creating educational courses in resource-limited settings where staff may not have as much time outside of clinical practice for training.

The best simulation model to use for these courses in LMICs remains unclear. Current literature suggested that cadaveric-based simulation sessions are the most beneficial method in teaching anatomy and operative skills [37–39]. All students are able to have similar experiences as procedures are easily repeatable on cadaveric models [21]. Conversely, other studies prefer the use of live animals as cadavers often have issues of tissue pliability as well as a lack of real-time feedback during bleeding and other physiological changes during the operative session [19]. Cost must also be taken into account, with Anderson et al. [19] stating that a live animal model was not feasible due to financial limitations. Artificial models may also be utilised for trauma simulation courses and offer a viable alternative to live animals, the use of which is becoming increasingly difficult due to ethical welfare societies. Studies have shown that these artificial models have better anatomic landmarks and positioning compared with live animal models [40,41]. A study by Garcia et al. [21] highlighted a clear preference of the TraumaMan model, but the significant $30,000 investment may be too large for most LMIC trauma units to purchase. There is a need for further research to develop and produce high-quality simulation models, such as SurgMan, affordably in order to provide participants with the best possible user experience.

Finally, further research in the form of prospective controlled trials is required to better answer our research question. Currently, there are limited published studies reporting patient outcomes following simulated trauma training courses [14]. A trial by Bellanova et al. [42] identified that individual surgeon and team training was associated with reduced mortality; however, multiple confounding variables greatly limits the strength of this finding. Studies should seek to identify the optimum course design for trauma surgery simulation courses in LMICs, including assessments of the student to instructor ratio, duration of the course, model of simulation and minimum requirement of resources. An assessment of surgeons' real-world performance following the course is also required to more accurately determine course efficacy.

## 5. Limitations

Limitations of this review are primarily due to the limited number of published studies in this area. Most of the included studies were non-randomised study designs with small sample sizes and did not account for confounding variables influencing outcomes. Only one study assessed long term outcomes [19], with the majority limited by a short follow-up time. It can also be argued that one of the studies, by Garcia et al. [21], is not representative of typical LMICs as the study took place in Sao Paulo, the richest state in Brazil, thus influencing the responses of participants when responding to survey questions.

Another significant limitation of this review is the lack of patient outcomes used to establish whether these courses have clinical benefit. As the aim of the courses is ultimately to improve patient care and mortality rates, it seems necessary to have patient outcomes at the forefront of this review. However, outcomes such as mortality are often influenced by a variety of external factors, both intra- and post-operatively, making it a suboptimal indicator when assessing the benefit of a trauma simulation course [43]. Furthermore, there is a paucity in the literature on the effect of any trauma-related educational initiatives on patient outcomes in LMICs; this is mainly due to insufficient patient record keeping in these regions.

Consequently, the outcomes reported in this review were mainly via participant test scores and self-reported surveys. Despite using pre- and post- test scores to produce an objective measure of knowledge improvement, these tests only reflect immediate knowledge recall of the participant rather than actual improvement in operative skill. A potential better measure of improvement in operative skill level can be via a practical exam assessed by course instructors; however, this presents a clear logistical and financial burden on

the training facility. In addition, any self-reported data are heavily subject to bias and are therefore difficult to use when forming conclusions on course benefit.

## 6. Conclusions

Our findings suggest that trauma surgery simulation courses in LMICs may improve surgeons' clinical knowledge and confidence in performing operative skills. However, this conclusion is drawn from a small number of studies with a small sample size and therefore caution must be exercised. In addition, the lack of universal performance metrics made the comparison between different courses challenging. Collaboration between international trauma institutions is essential for closing the educational resource inequality gap between higher- and lower-income countries. Further high-quality research is necessary to establish the optimum course design, and whether patient outcomes are improved following their implementation.

**Supplementary Materials:** The following are available online at https://www.mdpi.com/article/10.3390/traumacare1030012/s1, File S1: Search strategy results from Ovid Database, File S2: Full-text Exclusions (n = 72).

**Author Contributions:** Conceptualization, Y.H., Z.A.; methodology, Y.H., N.A.; formal analysis, Y.H., H.U., Z.A.; investigation, Y.H., H.U.; data curation, Y.H., A.J.; writing—original draft preparation, Y.H.; writing—review and editing, Y.H.; supervision, Z.A.; project administration, Y.H. All authors have read and agreed to the published version of the manuscript.

**Funding:** This research received no external funding.

**Institutional Review Board Statement:** Ethical review and approval were waived for this study as per advice from the NHS Health Research Authority (UK) decision tool, since it is a systematic review of published literature.

**Informed Consent Statement:** Patient consent was waived because no patients or members of the public were involved in the design and conduct of this study, or reporting of this research.

**Data Availability Statement:** All data generated as part of this study are included in the article.

**Conflicts of Interest:** The authors declare no conflict of interest.

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
