# Peer review of "Are Trauma Surgery Simulation Courses Beneficial in Low- and Middle-Income Countries—A Systematic Review and Meta-Analysis"

_traumacare, doi:10.3390/traumacare1030012_

Round 1

Reviewer 1 Report

Dear Authors, 

This  review was indeed a practical reflection of the challenges faced by many surgeons /trainees in LMIC's.

I have the following comments for your consideration:

  1. Was the protocol for this review registered  prior to commencement of this review? If yes, where?
  2. Line 20: the word 'utilising' does not refer to a purpose other than the intended with respect to the databases and should be changed.
  3. Line 114- 120: The primary and secondary outcomes should be clearly defined and placed earlier in the manuscript i.e. in the  background.
  4. Line 92: define what is meant by the 'benefit' of trauma simulation courses.
  5. Section 2.1: May include the three search strategies with the highest number of studies in the appendix for future reference.
  6. Section 2.2: Eligibility criteria can be tabulated to avoid repetition, transparent reporting and use the PICOST format.
  7. Section 2.3 : Data collection.  The content of this section are not ALL specific to data collection. It would be better to separate Study selection from data extraction
  8. Line 115: Comment pertaining to means / standard deviation should be placed in Section 2.5 . 
  9. Section 2.5 : Statistical analysis should be improved upon. Definitions of descriptive statistics as well as p value that is considered significant.
  10. Line 153: Spell all numbers less than ten in full throughout the document. '3' should be 'Three' as this is the first word of a sentence.
  11. Line 164: table 3 should be Table 3.
  12. Table 1: Can be improved
    1. Remove bullet points in sections
    2. Key for all abbreviated terms
    3. Check alignment 
  13. Table 2: Can be improved:
    1. Insert Key for all the abbreviated terms.
    2. Remove n numbers in the Outcome measures.
  14. Were the authors of the included studies contacted to provide clarifications for the extracted data?
  15. Line 348: The results of this study are not strong enough to support this statement. A statement of highlighting to the reader that the study findings are drawn from small sample size and are not of robust study design.
  16. Line 350.Move the sentence beginning with 'Further...' to the end of the paragraph.

Reviewer 2 Report

Thank you very much for this comprehensive review about two relevant topics as such as trauma knowledge and simulation.

The topics are furtherly demanding because the setting are low and middle income countries.

In my opinion non variations have to be done

Author Response

Thank you for the positive comments. No author responses are required.

Reviewer 3 Report

Thank you for the opportunity to review this interesting manuscript. It is an article reviewing whether the trauma surgery simulation courses are beneficial in low and middle-income countries. The authors are to be congratulated on their achievements.

In my opinion, the authors provided an interesting report. The objectives were clearly stated. The necessities of the analysis were adequately explained. The study method was adequately described. The results clearly presented. The discussion pointed out the important findings. The conclusions appropriately based on the results and discussions.

I am appreciated for the fact and congratulated on authors’ achievement. however, some concerns had raised from this work. Education and training for healthcare providers are the essential foundation for quality of care. Authors made lots of valuable discussions. However, none of the reviewed studies discussed the patient outcome related to simulation training. It is difficult for authors to answer the questions for part of secondary objectives of this study. Furthermore, in my point of view, the trauma system, including prehospital trauma care and the multidisciplinary trauma team, might have a major influence on patient outcome and are also critical for improving trauma patient care. I am appreciated if the authors may consider to add their comments into the discussion section.

Author Response

Author response: Thank you for positive comments.

Comment: I am appreciated for the fact and congratulated on authors’ achievement. however, some concerns had raised from this work. Education and training for healthcare providers are the essential foundation for quality of care. Authors made lots of valuable discussions. However, none of the reviewed studies discussed the patient outcome related to simulation training. It is difficult for authors to answer the questions for part of secondary objectives of this study. Furthermore, in my point of view, the trauma system, including prehospital trauma care and the multidisciplinary trauma team, might have a major influence on patient outcome and are also critical for improving trauma patient care. I am appreciated if the authors may consider to add their comments into the discussion section.

Author response: Further discussion as recommended by this reviewer is now included in Lines 475-479.

Round 2

Reviewer 1 Report

Dear Authors , 

Thank you for your responses and manuscript amendments.

I have the following comments for your consideration:

  1. Line 124-130: Data extraction should -describe the methodology of this step and is sufficiently described in line 118-123. The rationale for objective selection is best elaborated upon in Section 2.2 Eligibility Criteria.    Rationale for descriptive statistics  selection should be placed in Section 2.6 Statistical analysis.  To say that the primary outcome data is commonly presented as Mean( SD) is not a strong justification for selecting this descriptive statistic approach.  It appears that based on the small number of included studies, inferential statistics by way of meta-analysis is not possible . As a result a narrative synthesis of available data was undertaken.  This is still not made clear in the Section 2.6 Statistical Analysis and can be improved upon. 
  2. Line 174. These average score test should be more clearly annotated. What does 55.4 +- 13.9 ? Is this  a percentage ? Is this representing a mean (SD)?  Similarity for  78.1+- 11.6%.
  3. Table 1:  Study Design (Inclusion) - Primary data studies conducted in a High-income country as defined by the World Bank Classification   is not a study design. Here the reader expects the authors to list types of studies as completed for the Study Design -Exclusion criteria. The absence of comparator to the intervention can be interpreted in many ways: 1.this review is limited to observational studies only 2. this criterion was finalised after the search was completed. If this  was intended, then the potential selection bias for eligible studies should be noted in Section 5 Limitations.  Please improve the eligibility criteria definitions.
  4. Table 1: If only studies conducted in HIC  are included, why is this review evaluating surgical simulation in LMIC? Please clarify this  eligibility criterion.
  5. Table 2: What does (x3) refer to?
  6. Table 2: 6 didactic lectures can be improved to (n=6). This same caveat can be carried out in all tables for consistency.
  7. Table 2& 3: Complete the LEGEND 
  8. Table 4: Title can be improved to Test Scores ( Mean +- Standard Deviation)
  9. Please review the document again as there are still many areas where numbers less than ten are not spelt out in full. 
  10. For increased transparency, the list of excluded FULL TEXT ARTICLES (n=72) can be included as a supplementary file for the reader to access.

Author Response

Comment: Line 124-130: Data extraction should -describe the methodology of this step and is sufficiently described in line 118-123. The rationale for objective selection is best elaborated upon in Section 2.2 Eligibility Criteria.   

Author response: Done

Comment: Rationale for descriptive statistics selection should be placed in Section 2.6 Statistical analysis. 

Author response: Done.

Comment: To say that the primary outcome data is commonly presented as Mean (SD) is not a strong justification for selecting this descriptive statistic approach. 

Author response: We have specified that individual and combined mean scores were all analysed by descriptive statistics, as detailed in Table 4.

Comment: It appears that based on the small number of included studies, inferential statistics by way of meta-analysis is not possible. As a result, a narrative synthesis of available data was undertaken.  This is still not made clear in the Section 2.6 Statistical Analysis and can be improved upon. 

Response: Despite the small number of studies and participants, we were able to perform a meta-analysis which is now included as Figure 2. This showed a significant improvement in performance after taking part in a surgery simulation course. The title and abstract have been adjusted accordingly to reflect the meta-analysis.

Comment: Line 174. These average score test should be more clearly annotated. What does 55.4 +- 13.9? Is this a percentage? Is this representing a mean (SD)?  Similarity for 78.1+- 11.6%.

Response: Have changed lines 399-400 to clarify these values.

Comment: Table 1:  Study Design (Inclusion) - Primary data studies conducted in a High-income country as defined by the World Bank Classification   is not a study design. Here the reader expects the authors to list types of studies as completed for the Study Design -Exclusion criteria. The absence of comparator to the intervention can be interpreted in many ways: 1. this review is limited to observational studies only 2. this criterion was finalised after the search was completed. If this was intended, then the potential selection bias for eligible studies should be noted in Section 5 Limitations.  Please improve the eligibility criteria definitions.

Response: Eligibility criteria table 1 has been updated. The search strategy was not limited to only observational studies and was not finalised after the search was completed, therefore there is no selection bias to report in the limitations.

Comment: Table 1: If only studies conducted in HIC are included, why is this review evaluating surgical simulation in LMIC? Please clarify this eligibility criterion.

Response: Eligibility criterion has been updated

Comment: Table 2: What does (x3) refer to?

Response: This refers to three separate 60-minute sessions. The table has been improved to add clarity.

Comment: Table 2: 6 didactic lectures can be improved to (n=6). This same caveat can be carried out in all tables for consistency.

Response: Done

Comment: Table 2& 3: Complete the LEGEND 

Response: Legend has been completed in both tables.

Comment: Table 4: Title can be improved to Test Scores (Mean +- Standard Deviation)

Response: Title has been changed

Comment: Please review the document again as there are still many areas where numbers less than ten are not spelt out in full. 

Response: This has now been addressed

Comment: For increased transparency, the list of excluded FULL TEXT ARTICLES (n=72) can be included as a supplementary file for the reader to access.

Response: This has been added to supplementary files.